# Water-participated mild oxidation of ethane to acetaldehyde

Bin Li[1,2,9], Jiali Mu[1,9], Guifa Long[3,9], Xiangen Song[1] ✉, Ende Huang[1,2], Siyue Liu[1,2], Yao Wei[4], Fanfei Sun[4], Siquan Feng[1], Qiao Yuan[1,2], Yutong Cai[1,2], Jian Song[1,2], Wenrui Dong[5,6], Weiqing Zhang[5], Xueming Yang[5,7] ✉, Li Yan[1] & Yunjie Ding[1,8] ✉

The direct conversion of low alkane such as ethane into high-value-added chemicals has remained a great challenge since the development of natural gas utilization. Herein, we achieve an efficient one-step conversion of ethane to $C_2$ oxygenates on a $Rh_1$/AC-SNI catalyst under a mild condition, which delivers a turnover frequency as high as 158.5 $h^{-1}$. $^{18}O$ isotope-GC–MS shows that the formation of ethanol and acetaldehyde follows two distinct pathways, where oxygen and water directly participate in the formation of ethanol and acetaldehyde, respectively. In situ formed intermediate species of oxygen radicals, hydroxyl radicals, vinyl groups, and ethyl groups are captured by laser desorption ionization/time of flight mass spectrometer. Density functional theory calculation shows that the activation barrier of the rate-determining step for acetaldehyde formation is much lower than that of ethanol, leading to the higher selectivity of acetaldehyde in all the products.

With the growing demand for energy and chemical products in contemporary society, the utilization of natural gas has been attracting more and more attention. At present, most studies on the direct conversion of low-carbon alkanes at relatively lower temperatures are focused on methane[1–14]. Ethane ($C_2H_6$) is the second major component in shale gas, which features a high C–H bond energy of 423.29 kJ $mol^{-1}$ and thus high-temperature range of 700–1000 K is usually needed for its cracking and dehydrogenation into ethylene[15–20]. However, the high-temperature process often brings about significant side reactions, such as coke or deactivation of the catalyst, and huge energy consumption. Therefore, it is of great significance to develop a direct conversion path of ethane under lower temperature conditions. There have been not so many reports on the direct conversion of ethane under mild conditions[2–4,21–24]. Hutchings et al. found that ethane could be directly converted to ethanol and ethyl hydroperoxide, which

would be further oxidized to acetaldehyde and acetic acid at 323 K on Fe-ZSM-5 catalyst using $H_2O_2$[21]. Martin et al. realized low-temperature oxidation of ethane to oxygenates by oxygen over Ir-cluster catalysts in the presence of CO[24]. Furthermore, the indispensable role of CO in the reaction system has also been studied[25,26]. Nevertheless, there are still two major challenges in the current field of direct oxidation of alkanes: (a) to study whether the active sites provided by single-atom catalysts are different from the pathways used by traditional catalysts to activate alkanes and (b) to figure out the cooperative oxidation mechanism of $H_2O$ and $O_2$ in alkane activation.

Heterogeneous single-metal-site catalysts (HSMSCs) have recently emerged as an important class of high-efficiency catalysts with almost 100% atomic utilization and unique properties[27,28], which can ideally bridge the gap between heterogeneous and homogeneous catalysis and offer a platform for understanding the nature of active

[1]Dalian National Laboratory for Clean Energy, Dalian Institute of Chemical Physics, Chinese Academy of Sciences, Dalian, China. [2]University of Chinese Academy of Sciences, Beijing, China. [3]Guangxi Key Laboratory of Chemistry and Engineering of Forest Products, School of Chemistry and Chemical Engineering, Guangxi Minzu University, Nanning, China. [4]Shanghai Synchrotron Radiation Facility, Shanghai Advanced Research Institute, Chinese Academy of Sciences, Shanghai, China. [5]State Key Laboratory of Molecular Reaction Dynamics, Dalian Institute of Chemical Physics, Chinese Academy of Sciences, Dalian, China. [6]Hefei National Laboratory, Hefei, China. [7]Department of Chemistry, Southern University of Science and Technology, Shenzhen, China. [8]State Key Laboratory of Catalysis, Dalian Institute of Chemical Physics, Chinese Academy of Sciences, Dalian, China. [9]These authors contributed equally: Bin Li, Jiali Mu, Guifa Long. ✉e-mail: xiangensong@dicp.ac.cn; xmyang@dicp.ac.cn; dyj@dicp.ac.cn

sites at the molecular level[29]. The relationships between coordination atoms and metal centers determine the electronic state and geometry of a single-metal site and further the performance of catalytic reactions. In recent years, the gradual development of tuning the local coordination environment of single-metal-sites by doping N atoms on carbon carriers has emerged[30,31]. Additionally, sulfur as an electron-donating ligand can also form a coordination bond with the active center, influencing electron cloud density and thus enhancing reaction activity[32,33]. We previously revealed that S species can promote the activity of methanol carbonylation by coordination with single-Rh-site[34,35]. Furthermore, the crucial role of I species in achieving atomic dispersion of Rh nanoparticles (NPs) was investigated[27,36]. Introducing multifarious coordination atoms (S, N, and I) onto the active carbon support as anchoring sites to immobile single-metal-site offers a prospective pathway for synthesizing HSMSCs.

Herein, we first show that the single-Rh-site bound on S, N, and I-doped activated carbon (Rh$_1$/AC-SNI) in the form of Rh mononuclear complex efficiently catalyzes the direct conversion of ethane to C$_2$ oxygenate products using an O$_2$ oxidizing agent at 423 K in aqueous solution. Moreover, the turnover frequency (TOF) of the oxygenate products can reach as high as 158.5 h$^{-1}$ on Rh$_1$/AC-SNI, which is the highest activity compared with the reported results in the literature so far. The ethane activation on the Rh$_1$ was systematically investigated by a variety of characterizations, which revealed that O$_2$ and H$_2$O could directly participate in the reaction of ethane oxidation. Isotope-GC−MS results show that the formation of ethanol and acetaldehyde follows two different reaction pathways, and the O source of ethanol product originates from O$_2$, while H$_2$O directly involved in the generation process of acetaldehyde. Experimental results combined with density functional theoretical (DFT) calculations show that the energy barrier for the production of ethanol at the Rh active site is higher than the energy barrier for the production of acetaldehyde.

## Results

### Catalyst characterization

The Rh$_1$/AC-SNI catalyst was synthesized based on the strong coordination interaction between a single Rh ion and N, S, and I ligands on the AC (see details in the Experimental section). As shown in Fig. 1a, activated carbon (AC) was first co-doped with S and N, followed by I incorporation. Subsequently, the modified AC was coordinated with Rh at ambient conditions to get the final product. For comparison, the samples with single N or S doping were prepared using the same protocol, which is denoted as Rh$_1$/AC-SI and Rh$_1$/AC-NI, respectively. We explored the excellent microstructural properties of Rh$_1$/AC-SNI and comparative catalysts by scanning electron microscopy (SEM), nitrogen adsorption-desorption, thermogravimetric, and Raman spectroscopy experiments (Supplementary Figs. 1–4 and Supplementary Table 1)[37,38]. In line with the X-ray diffraction (XRD) results (Supplementary Fig. 5), no clusters or NPs were observed by the high-resolution transmission electron microscopy (HR-TEM) (Fig. 1b and Supplementary Fig. 6). The aberration-corrected high-angle annular dark-field scanning transmission electron microscopy (AC HAADF-STEM) image further indicates the atomic dispersion of Rh ions in the carbon matrix according to the prominent Z-contrast difference between Rh and C atoms and the absence of Rh NPs or clusters (Fig. 1c and Supplementary Fig. 7). Energy dispersive spectroscopy (EDS) mapping confirms the uniform distribution of S, N, I, and Rh on Rh$_1$/AC-SNI (Fig. 1d). Similar structural and morphological information was validated on the Rh$_1$/AC-SI and Rh$_1$/AC-NI catalysts (Supplementary Figs. 1–7).

The X-ray absorption near-edge structure (XANES) spectra exhibit that all the absorption of Rh on Rh$_1$/AC-NI, Rh$_1$/AC-SI, and Rh$_1$/AC-SNI is located between those of RhI$_3$ and Rh foil, revealing that the valence state of Rh$^{\delta+}$ species is between 0 and +3 (Fig. 1e). Based on Fig. 1f, g, Rh−Rh (8.9 Å$^{-1}$) and Rh-I (9.6 Å$^{-1}$) can be distinguished, further verifying

the absence of Rh NPs. Subsequently, the detailed extended X-ray absorption fine structure (EXAFS) fitting was applied to extract quantitative structural results for the Rh moiety in Rh$_1$/AC-SNI. The peaks at 2.35 and 2.66 Å correspond to Rh-S and Rh-I coordination in the fresh Rh$_1$/AC-SNI, with coordination numbers of 0.7 and 2.6, respectively. The Rh−C/N bonds are assigned as 2.07 Å in the process of fitting, and their coordination number is 1.4 (Fig. 1f, Supplementary Fig. 8, and Supplementary Table 2). To build a rational structure of the catalyst, we performed CO temperature-programmed desorption mass spectrometry (CO-TPD MS) experiments. It can be found that the molar ratio of CO/Rh for Rh$_1$/AC-SNI, Rh$_1$/AC-NI, and Rh$_1$/AC-SI are 1.06, 1.04, and 1.94, respectively, suggesting the coordination number of Rh-CO on Rh$_1$/AC-SNI, Rh$_1$/AC-NI, and Rh$_1$/AC-SI were 1, 1, and 2, respectively[39,40] (Supplementary Fig. 9). Combined results from XAFS and CO-TPD MS, we can conclude that the Rh$_1$ on Rh$_1$/AC-SNI mainly exists in the form of Rh(CO)I$_3$(N-AC)(S-AC). S-AC and N-AC denoted the sulfur and nitrogen-containing groups. Similarly, Rh(CO)$_2$I$_2$(S-AC) (O-AC) and Rh(CO)I$_3$(N-AC)(O-AC) are the main molecular structure of Rh$_1$/AC-SI and Rh$_1$/AC-NI, respectively. In accordance with the XANES results, the X-ray photoelectron spectroscopy (XPS) of Rh 3$d$ shows the coexistence of Rh$^{1+}$ and Rh$^{3+}$ species on Rh$_1$/AC-SNI, Rh$_1$/AC-NI, and Rh$_1$/AC-SI catalysts (Supplementary Fig. 10 and Supplementary Table 3). Additionally, Maria and Ma Ding et al. believe that CO is an important ligand that can maintain the valence state of metals[6,24]. We designed quasi-in situ XPS to analyze the role of CO on Rh$_1$/AC-SNI during the reaction. The sample was pretreated in a mixture of C$_2$H$_6$, H$_2$O, and O$_2$ at 423 K for 2 h. The binding energy of Rh 3$d$ 5/2 overall shifted to a higher value at 311.6 eV (Supplementary Fig. 11 and Supplementary Table 3). This indicates that CO plays a role in maintaining the valence state of Rh during the reaction, ensuring the optimal coordination environment of active sites. As shown in Supplementary Figs. 12–13 and Supplementary Table 4, comparing with the N species on the AC-SNI carrier, it was found that a new peak appeared at 399.2 eV after Rh loaded on Rh$_1$/AC-SNI, confirming the presence of Rh−N coordination[41–43]. Simultaneously, the binding energy of C−S−C at fresh Rh$_1$/AC-SNI shifted to low energy direction about 0.4 eV (Supplementary Fig. 14 and Supplementary Table 5), in contrast to that of AC-SNI, suggesting that the S of the C−S−C species withdrew electrons from Rh, causing increase of the electron cloud density around S. Furthermore, we calculated the Bard charges of Rh on Rh$_1$/AC-SNI, Rh$_1$/AC-SI, and Rh$_1$/AC-NI catalysts and found that the chemical state of Rh$_1$/AC-SNI was the highest (Supplementary Fig. 15), consistent with the XPS and XANES experimental results.

### Catalyst evaluation

A typical experiment was carried out in a 100 mL batch reactor containing a polytetrafluoroethylene (PTFE) lining by adding 50 mg catalyst to 10 g deionized water under the conditions of 30 bar C$_2$H$_6$, 1 bar CO, and 0.5 bar O$_2$ at 423 K. The catalytic performance for the oxidation of ethane was measured over these catalysts. For comparison, the Rh$_{NPs}$/AC catalyst results in a higher activity than those of Rh/SiO$_2$ and Rh/CeO$_2$ (Supplementary Figs. 16 and 17). Nonetheless, it is still inferior to that of the series of Rh$_1$/AC-x catalysts (Fig. 2a). Moreover, the activity test shows that Rh$_1$/AC-SNI is superior to that Rh$_1$/AC-SN, Rh$_1$/AC-I, Rh$_1$/AC-SI, and Rh$_1$/AC-NI under the same condition, demonstrating the positive effect of nitrogen and sulfur species on ethane oxidation (Fig. 2a and Supplementary Table 6).

With the increase of the partial pressure of C$_2$H$_6$ from atmospheric pressure to 3.0 MPa in Fig. 2b, the TOF of total C$_2$ oxygenated products steadily increased. Along with the partial pressure of CO increase from atmospheric pressure to 1.5 MPa, the TOF of total C$_2$ oxygenates products reached the maximum value of 158.5 h$^{-1}$ and the ethane conversion is 1.98% at 1.25 MPa (Fig. 2c). An excess of CO concertation will reduce the activity. Similarly, when the partial pressure of O$_2$ is 0.25 MPa, the TOF of oxygenated compounds

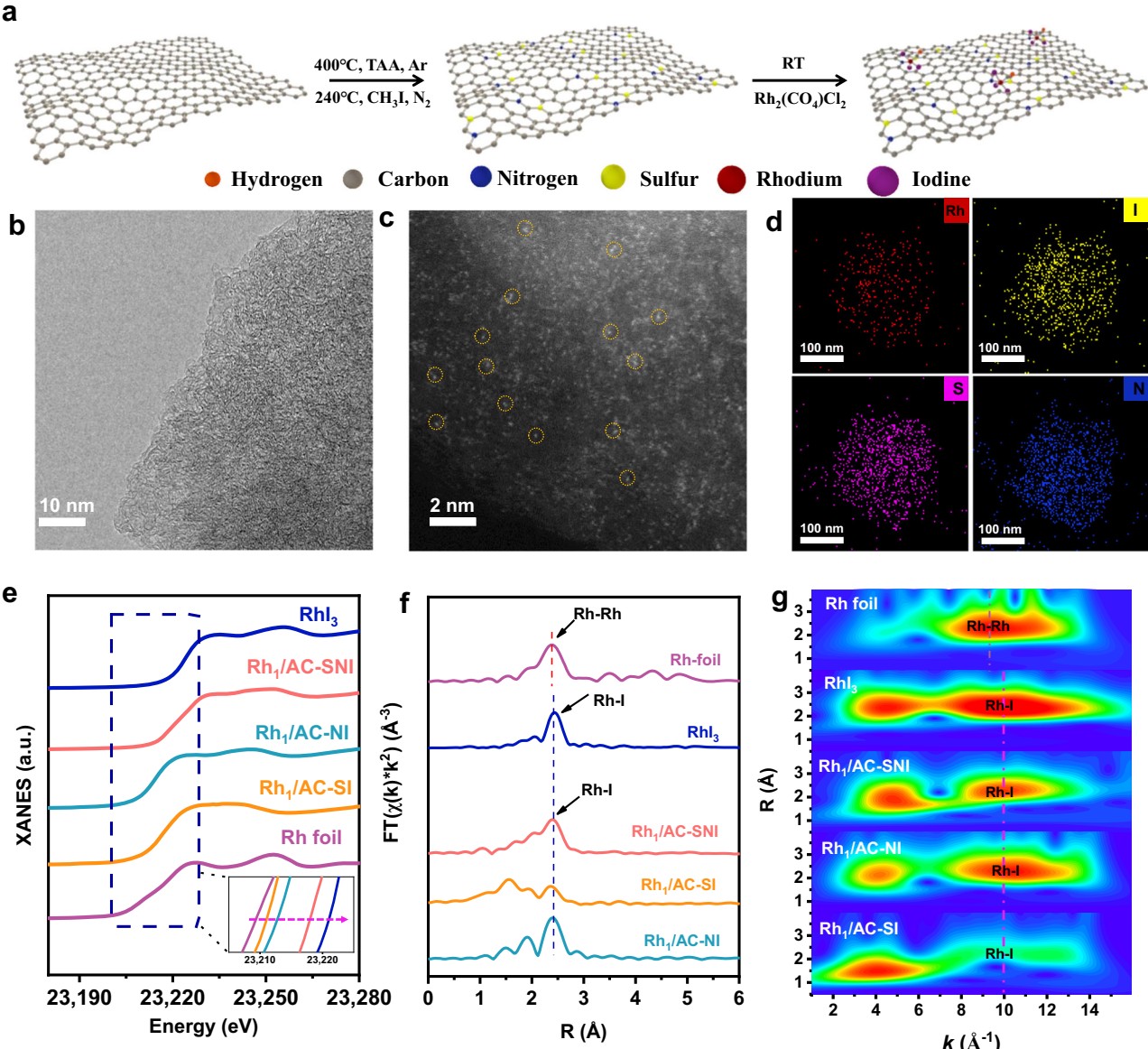

**Fig. 1 | Synthesis and structural characterization of samples. a** Schematic illustration of the preparation of Rh$_1$/AC-NI, Rh$_1$/AC-SI, and Rh$_1$/AC-SNI catalysts. **b** HR-TEM images of fresh Rh$_1$/AC-SNI. **c** HAADF-STEM images of fresh Rh$_1$/AC-SNI. **d** HAADF-EDS-mapping of fresh Rh$_1$/AC-SNI. **e** XANES spectra for Rh foil, RhI$_3$, Rh$_1$/ AC-NI, Rh$_1$/AC-SI, and Rh$_1$/AC-SNI. **f** The experimental curve of $k^2$-weight EXAFS spectra in R-space of Rh foil, RhI$_3$, Rh$_1$/AC-NI, Rh$_1$/AC-SI, and Rh$_1$/AC-SNI. **g** The wavelet transforms contour plots of $k^2$-weighted $\chi(k)$ EXAFS signals of Rh foil, RhI$_3$, Rh$_1$/AC-NI, Rh$_1$/AC-SI, and Rh$_1$/AC-SNI.

reaches a maximum (Fig. 2d), which indicates that higher oxygen pressure will cause over-oxidation of the product to CO$_2$ and keeping the ratio of CO/O$_2$ greater than 2.0 will result in a higher TOF. By performing parallel experiments to remove any one of C$_2$H$_6$, CO, or O$_2$, none of the studies showed the production of ethanol, acetaldehyde, acetic acid, and other products, suggesting that the participation of all three gases is indispensable for the formation of C$_2$ oxygenates. When the temperature increased from 373 to 453 K, the conversion of ethane and TOF of total C$_2$ oxygenates products both increased significantly. As the main product, the selectivity of acetaldehyde remains basically unchanged at about 70%. The TOF of Rh$_1$/AC-SNI can reach 34.5 h$^{-1}$ at 373 K (Fig. 2e), greatly exceeding 7.5 h$^{-1}$ on Ir-based catalysts by Martin[24] under the same temperature condition. With the prolonged reaction time, the TOF of total C$_2$ oxygenates products also increased until 8 h and then remained unchanged (Fig. 2f). The reaction order of ethane to generate ethanol on Rh$_1$/AC-SNI is 0.6 with the reaction activation energy of 32.99 kJ mol$^{-1}$, and the reaction order of generating acetaldehyde is

0.7 with the reaction activation energy of 30.22 kJ mol$^{-1}$, which is consistent with the higher acetaldehyde selectivity (Fig. 2g). In addition, the stability of Rh$_1$/AC-SNI catalyst was verified with seven reaction cycles, showing ignorable activity decay at 423 K, 4.5 MPa total pressure and 2 h reaction time (Fig. 2h). It is worth noting, that the highest TOF of C$_2$ oxygenates on Rh$_1$/AC-SNI came up to 158.5 h$^{-1}$ at 423 K, higher than those of previous reports on direct conversion of ethane at low-temperature[21,24]. Meanwhile, the activity of Rh$_1$/AC-SNI was also obviously higher than those of Mo, Pt, Pd, and Ir supported on AC-SNI with the same loading, the highest activity of which is only one-sixth of Rh$_1$/AC-SNI under the same reaction conditions (Supplementary Fig. 18 and Supplementary Table 7). In addition, it is found that the microstructure of the Rh$_1$/AC-SNI-spent catalyst did not change significantly after the reaction (Supplementary Figs. 2e, 6d, and 7d) and the coordination environment remained basically the same comparing the AC-HADDF-STEM and EXAFS characterization results (Supplementary Figs. 19–21), indicating the stable coordination ability of Rh$_1$/AC-SNI.

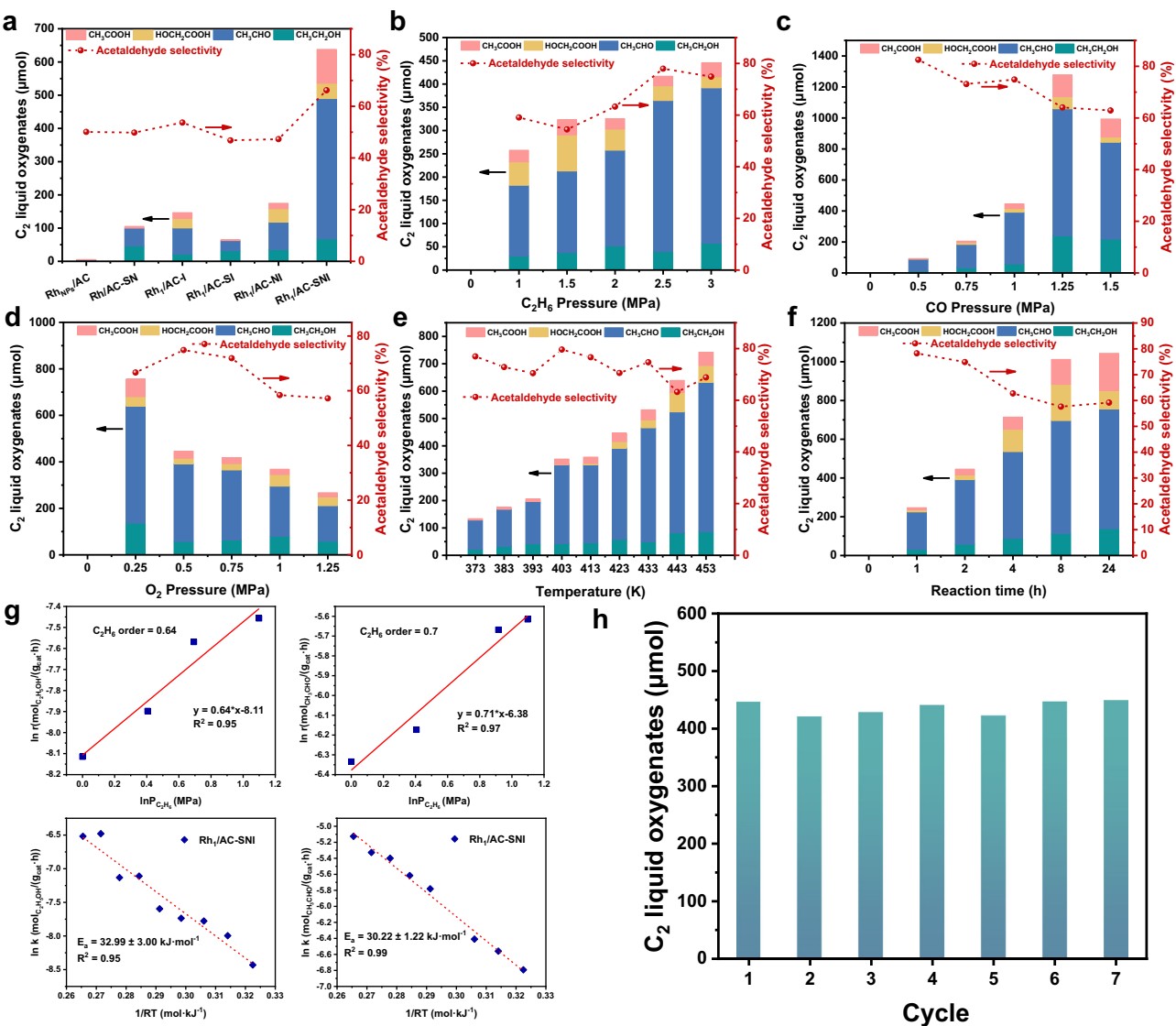

**Fig. 2 | Ethane oxidation performance and structure-activity relationship.** Catalytic activity of **a** different catalysts. **b** Different pressure of C₂H₆. **c** Different pressure of CO. **d** Different pressure of O₂. **e** Different reaction temperatures. **f** Different reaction time. **g** Reaction kinetic test on Rh₁/AC-SNI for ethane direct oxidation. Glycolic acid is included in the ethanol pathway and acetic acid is included in the acetaldehyde pathway. **h** Recyclability tests of Rh₁/AC-SNI (general reaction conditions: $T = 423\,K$, $P_{C_2H_6} = 3.0\,MPa$, $P_{CO} = 1.0\,MPa$, and $P_{O_2} = 0.5\,MPa$ over Rh₁/AC-SNI for 2 h, $m\,(H_2O) = 10\,g$, $m\,(catalyst) = 50\,mg$).

For the isotope labeling experiments on Rh₁/AC-SNI catalyst in Fig. 3a–c, the mass fragment groups of ethanol, acetaldehyde, and acetic acid were CH₃O and C₂H₅O, CHO, C₂H₃O, and C₂H₄O, C₂H₃O, CHO₂ as well as C₂H₄O₂ species (Supplementary Tables 8–10). Only one D-atom exchange occurred in ethanol product (Fig. 3a), while multiple D-atoms exchange was found in acetaldehyde (Fig. 3b, c), suggesting that the ethyl group only participated in the formation of ethanol while vinyl or its further dehydrogenation intermediates were involved in the formation of acetaldehyde. To identify the carbon source of C₂ oxygenates, the ¹³CO isotope labeling experiment was carried out on the Rh₁/AC-SNI catalyst. The signal of ¹³CO₂ (125.25 ppm) could be clearly observed. In contrast, no signal of ¹³C appeared in C₂ oxygenates (Fig. 3d), indicating that CO did not directly participate in the reaction but worked as a co-catalyst. And C₂H₆ was the only carbon source of the C₂ oxygenates. To verify the intermediate species of ethane dehydrogenation, we conducted the C₂H₆–D₂-TPD-MS pulse experiment on Rh₁/AC-SNI (Fig. 3e). A series of C₂H₆–D₂ exchange products on the Rh₁/AC-SNI catalyst at 423 K, and C₂H₅D ($m/z = 31$), C₂H₄D₂ ($m/z = 32$), C₂H₃D₃ ($m/z = 33$), C₂H₂D₄ ($m/z = 34$), C₂HD₅

($m/z = 35$), and C₂D₆ ($m/z = 36$) can be observed. Since the mass spectrometer uses an electron impact ionization source (EI source) to detect the intermediates, the signal at $m/z = 31$ contains a part of species aroused by ethane isotope natural abundance. Nevertheless, it can be observed that the intensity of $m/z = 31$ species quickly reinforced as the number increase of ethane pulse as shown in Supplementary Fig. 22, indicating the formation of C₂H₅D species except for the trace natural isotope. In addition, the signal increase of C₂H₄D₂ among these species is the most remarkable with the increase of ethane pulse times (Fig. 3e). These results indicated that multiple C–H bonds of ethane can be simultaneously activated on Rh₁/AC-SNI and dissociated into different dehydrogenation species, participating in different reaction pathways, respectively.

We also performed isotopic experiments using ¹⁸O-labeled oxygen and analyzed the liquid-phase product for the Rh₁/AC-SNI catalyst. A large amount of C₂H₅¹⁸O ($m/z = 47$) was generated, showing that ethanol product was basically marked with ¹⁸O (Fig. 3a). Surprisingly, almost none of ¹⁸O was detected in acetaldehyde (Fig. 3b, c), indicating that O₂ only participated in the formation of ethanol. To further

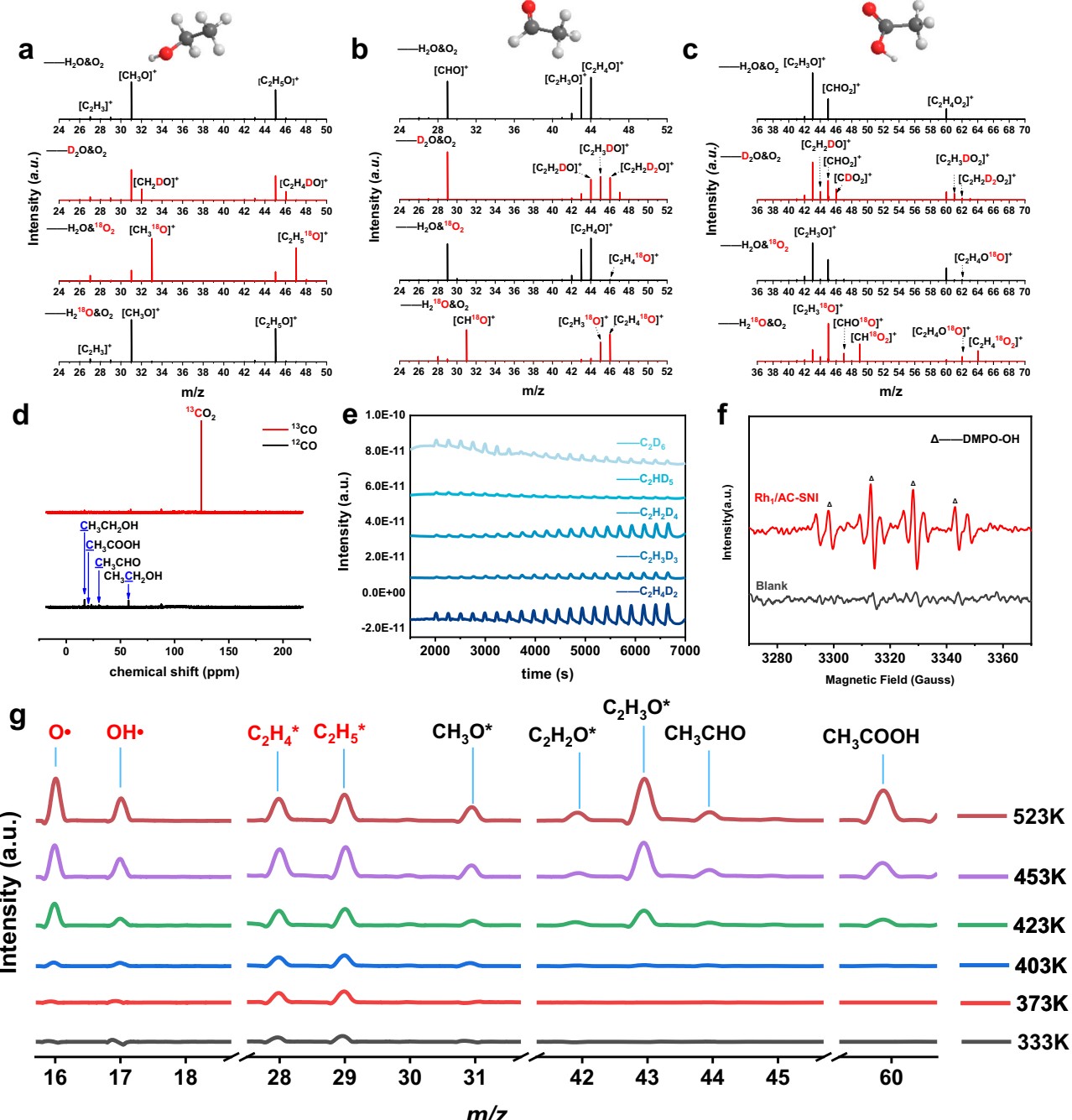

**Fig. 3 | In situ experiments. a** Ethanol, **b** acetaldehyde, and **c** acetic acid GC–MS spectra, respectively with $H_2O/D_2O/^{18}O_2/H_2^{18}O$ isotope labeled on $Rh_1/AC$-SNI catalyst. **d** Liquid-phase $^{13}C$ NMR product spectra of CO and $^{13}CO$ as reactants. **e** $C_2H_6–D_2$-TPD-MS experiment on $Rh_1/AC$-SNI catalyst. **f** EPR spectra of the ethane oxidation reaction over $Rh_1/AC$-SNI. DMPO was added to the reaction mixture as the radical trapping agent. **g** In situ FEL-TOF/MS spectrometry on $Rh_1/AC$-SNI using $C_2H_6/CO/O_2/H_2O/Ar$ at 303–523 K.

explore the oxygen sources of acetaldehyde, we used $^{18}O$-labeled water for isotope experiments (Supplementary Tables 8–10). Figure 3a shows that none of the ethanol was labeled by $^{18}O$ from water. While all $CH^{18}O$ ($m/z = 31$), $C_2H_3^{18}O$ ($m/z = 45$), and $C_2H_4^{18}O$ ($m/z = 46$) fragments from acetaldehyde were labeled by $^{18}O$ from water (Fig. 3b). As shown in Fig. 3c, most of the O in acetic acid originated from $H_2^{18}O$, verified by a large amount of $CH^{18}O_2$ ($m/z = 49$) and $C_2H_4^{18}O_2$ ($m/z = 64$). Meanwhile, a small amount of oxygen in acetic acid also came from $O_2$, forming $CHO^{18}O$ ($m/z = 47$) and $C_2H_4O^{18}O$ ($m/z = 62$), indicating that only a small part of acetic acid derived from the oxidation of ethanol by $O_2$. Additionally, we explored the universality of the reaction

mechanism using Rh-ZSM-5 catalyst. The catalytic activity and $^{18}O$-GC–MS results are shown in Supplementary Figs. 23 and 24. Despite of the higher acetic acid selectivity of Rh-ZSM-5, the $^{18}O$ distribution of oxygenates for Rh-ZSM-5 is basically consistent with that of $Rh_1/AC$-SNI, indicating that the low-temperature direct oxidation reaction mechanism of ethane proposed here has a certain universality. Oxygen exchange between $H_2O$ and $O_2$ is negligible (Supplementary Fig. 25)[44–48]. Isotopic experiments indicated that $H_2O$ directly participated in the reaction process and provided an oxygen source for acetaldehyde. To explore the reaction routes, ethanol and acetaldehyde were used as a substrate to substitute $C_2H_6$, respectively. The

formation rate of acetaldehyde and acetic acid was very slow using ethanol substrate. However, acetaldehyde substrate was quickly converted into acetic acid, suggesting that the oxidation energy barrier of ethanol was much higher than that of acetaldehyde to form acetic acid on $Rh_1$/AC-SNI catalyst (Supplementary Fig. 26). Electron paramagnetic resonance (EPR) with 5,5-dimethyl-1-pyrroline N-oxide (DMPO) as the radical scavenger was conducted[45,49]. Figure 3f showed that hydroxyl radicals (•OH) decomposed from $H_2O$ during the reaction process can be captured by DMPO in the liquid phase with clear response signals, which directly participated in the reaction with ethyl or ethylidene on $Rh_1$/AC-SNI catalyst.

In situ free-electron laser time of flight mass spectrometry (FEL-TOF/MS) using the vacuum ultraviolet free-electron laser (VUV FEL) at the Dalian Coherent Light Source of China (Supplementary Fig. 27)[50,51]. Some key intermediate species, such as oxygen radicals (•O), hydroxyl radicals (•OH), vinyl groups ($C_2H_4$*), and ethyl groups ($C_2H_5$*) were captured when $C_2H_6$/CO/$O_2$/$H_2O$/Ar mixture passed through $Rh_1$/AC-SNI (Fig. 3g and Supplementary Fig. 28). Meanwhile, we also observed the obvious signals of $CH_3O$*, $C_2H_2O$*, $C_2H_3O$*, acetaldehyde, acetic acid, and other products and their corresponding fragmentation species at 523 K. The signal intensity of the intermediate species gradually strengthened with the temperature increase. The signals of •O ($m/z = 16$), •OH ($m/z = 17$), acetaldehyde, and acetic acid started to be obvious at 423 K, showing that the splitting of oxygen and water in the reaction system needs a minimum activation temperature. It can be concluded that oxygen and water both provided an O source for $C_2$ oxygenates in the reaction. Oxygen and ethyl groups from ethane played a determining role in the formation of ethanol. Correspondingly, water and vinyl intermediates decided the generation of acetaldehyde. The generation pathways of ethanol and acetaldehyde should follow two different pathways, also proposed by Pan et al. in the process of ethane oxidation by in situ synchrotron radiation photoionization mass spectrometry[51].

Density functional theory (DFT) calculations were used to simulate the reaction path and energy change. In the path of ethanol formation (Fig. 4a, b), the O radical from $O_2$ would replace two I atoms in the initial $Rh_1(N-AC)(S-AC)(CO)I_3$ (**A1**) species and generate the transition state $Rh_1–2O$* (**TS1**) with an activation barrier of 1.35 eV as the rate-determining step. $Rh_1–2O$* can activate $C_2H_6$ to $C_2H_5$* and generate •OH to form $Rh_1–(O$*) $(OH$*)$(C_2H_5$*) (**TS2**) with the attack of ethane. Then the ethyl group migrates from the $Rh_1–(O$*)$(OH$*) active site to the hydroxyl group with an energy barrier of 0.21 eV to form $Rh_1–(O$*) $(CH_3CH_2OH$*) (**TS3**). Ethanol species are rapidly released into the solution without any energy barrier. Interestingly, when the •OH provided by $H_2O$ is used as the active O species, the energy barrier for ethane to break the C–H bond is as high as 3.33 eV (Supplementary Fig. 29-TS2), which indicates that $O_2$ can activate ethane molecules more effectively in the ethanol formation pathway. Philippe et al. found that alkane-to-metal donation determines the stability of the metal-alkane complex and metal-to-alkane back-donation facilitates C–H bond cleavage by oxidative addition in the σ-complexes[52]. Herein, we activated the C–H bond by constructing similar σ-complexes and further transformed it into different oxygenates.

For the path of acetaldehyde generation (Fig. 4c, d), the $H_2O$ molecule also replaced two I atoms of the same initial active site $Rh_1(N-AC)(S-AC)(CO)I_3$ (**C1**) to sequentially form $Rh_1–(OH$*) (**C2**) via the transition state $Rh_1–H_2O$ (**TS1**). **C2** was further oxidized by $O_2$ to form $Rh_1–O$* (**C3**). Ethane near $Rh_1–O$* will be activated to $C_2H_5$ and adsorbed on the $Rh_1$ active site to form $Rh_1–OH$*–$C_2H_5$*(**TS2**) as the rate-determining step with an activation barrier of 0.74 eV. $C_2H_5$* will be oxidized to vinyl by active •O in water and instantly bond with OH* via the **TS3** transition state to form $Rh–C_2H_4OH$*(**C7**). $C_2H_4OH$* is extremely prone to reconstitution. Then the formed acetaldehyde detaches

from the Rh active site. Nevertheless, the energy barrier of the rate-determining step is as high as 1.35 eV (Supplementary Fig. 30-TS1) when oxygen participates in the formation pathway of acetaldehyde, indicating that $H_2O$ is a more efficient oxidant in the formation pathway of acetaldehyde. Moreover, the energy barrier of the rate-determining step for acetaldehyde formation was 0.74 eV, much lower than 1.35 eV of ethanol formation, consistent with the higher selectivity of acetaldehyde in all the products. The Bader charge shows that the electron cloud density around O gradually increases in both paths (**A1-TS2** and **C1-TS2**), and the nucleophilicity of O increases, which is conducive to capturing H on $C_2H_6$ (Fig. 5). Before O captures H, the electron cloud around Rh gradually decreases, while the Rh–O bond interaction becomes weaker, and the electron cloud density of Rh increases after the σ-bond in O–H is formed, which is still conducive to the nucleophilic attack of the ethyl group.

We also compared the energy barriers of ethane with $O_2$ to ethanol over the $Rh_1$/AC-SI and $Rh_1$/AC-NI catalysts as shown in Supplementary Figs. 31 and 32. The energy barrier of the rate-determining step follows in order of $Rh_1$/AC-SI > $Rh_1$/AC-NI > $Rh_1$/AC-SNI (Supplementary Fig. 33 and Supplementary Tables 11, 15, and 16), which is also consistent with their corresponding catalytic activity (Fig. 2a).

## Discussion

In summary, we have successfully synthesized a series of $Rh_1$ catalysts supported on S, N, and I co-doped AC. The coordination environment of $Rh_1$ was modulated by the functional groups in the carbon matrix, as proved by various experimental characterizations. Among these catalysts, $Rh_1$/AC-SNI exhibited the highest $C_2$ oxygenates TOF of 158.5 $h^{-1}$ for the selective oxidation of ethane at low-temperature, obviously higher than those of $Rh_1$/AC-NI, $Rh_1$/AC-SI, $Rh_1$/AC-SN, $Rh_1$ /AC-I, and $Rh_{NPs}$/AC catalysts under the same mild reaction conditions. Acetaldehyde was the main product in all the $C_2$ oxygenates. The formation mechanism of acetaldehyde and ethanol followed two independent pathways, verified by an $^{18}O$ isotope labeling experiment. The oxygen source of ethanol and acetaldehyde is derived from oxygen and water, respectively. Oxygen radicals (•O), hydroxyl radicals (•OH), vinyl groups ($C_2H_4$*), and ethyl groups ($C_2H_5$*) key intermediates were in situ captured by in situ FEL-TOF/MS. Density function theoretical calculations also demonstrated the different reaction pathways for the generation of ethanol and acetaldehyde on $Rh_1$/AC-SNI catalyst, showing that the energy barrier of the rate-determining step for acetaldehyde production was 0.74 eV, lower than 1.35 eV of ethanol formation.

## Methods

### Preparation of S, N-doped activated carbon (AC/SN)

For the synthesis of AC-SN, 1.0 g AC, and 1.0 g thioacetamide ($C_2H_5NS$) were mixed well, then the uniformly mixed powder was heated to 673 K with a heating rate of 1 K $min^{-1}$ under a flowing Ar (30 mL $min^{-1}$) atmosphere for 2 h. After cooling to room temperature, the resulting samples were washed with ultra-pure water three times and then dried at 393 K to get the AC-SN support. For the preparation of N-doped activated carbon (AC-N) and S-doped activated carbon (AC-S), the synthesis protocol was similar to that of the AC-SN except that the precursor melamine ($C_3H_6N_6$) and benzyl disulfide ($C_{14}H_{14}S_2$) were used as the N and S sources, respectively.

### Preparation of catalysts

The AC-SN was pretreated with $CH_3I$ in a quartz tube furnace at 513 K for 4 h, and $CH_3I$ was bubbled by $N_2$ flow (50 mL $min^{-1}$) into the tube furnace. The as-prepared sample was denoted as AC-SNI. For comparison, pure AC, AC-N, and AC-S samples were treated with the same procedure, denoted as AC-I, AC-NI, and AC-SI, respectively. Subsequently, a certain amount of $Rh_2(CO)_4Cl_2$ was dissolved in $CH_2Cl_2$ (15 mL), and the 1.0 g AC-SNI was added into the solution for 24 h stirring at room temperature. The final product was filtered and washed with $CH_2Cl_2$ at least three

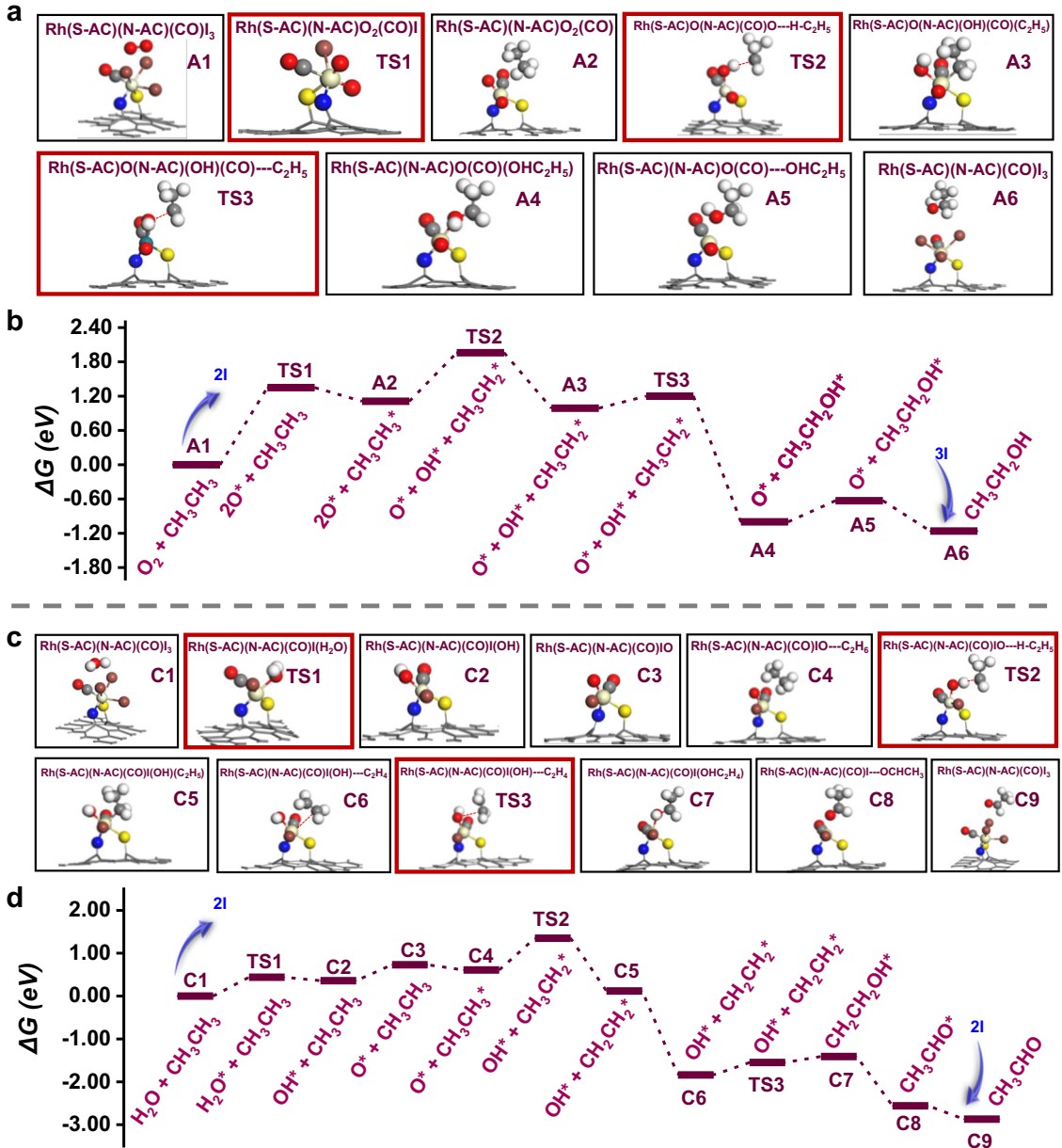

**Fig. 4 | DFT calculations. a** Structures of the key intermediates involved in the reaction pathway of $C_2H_6$ and $O_2$ to ethanol on $Rh_1$/AC-SNI catalyst. **b** The free energy ($\Delta G$) diagrams of the reaction pathway of $O_2$ participating in the production of ethanol. The states A1–A6 represent different basic states in the reaction pathway, and TS represents the transition state. **c** Structures of key intermediates involved in the reaction pathway of $C_2H_6$ and $H_2O$ to acetaldehyde on $Rh_1$/AC-SNI catalyst. **d** The free energy ($\Delta G$) diagrams of the reaction pathway of $H_2O$ participating in the production of acetaldehyde. States C1–C9 represent different basic states in the reaction pathway, and TS represents the transition state. Colors in the picture: the white balls are H; the gray balls are C; the red balls are O; the blue balls are N; the yellow balls are S; the brown balls are I; the beige balls are Rh.

times and dried at 333 K overnight to get the $Rh_1$/AC-SNI catalyst. Similarly, $Rh_1$/AC-I, $Rh_1$/AC-SN, $Rh_1$/AC-NI, and $Rh_1$/AC-SI samples were also prepared with the same method using AC-I, AC-SN, AC-NI, and AC-SI supports. $Rh/SiO_2$ and $Rh/CeO_2$ were prepared by traditional impregnation method using $RhCl_3$ aqueous solution and then calcined at 673 K for 2 h under Ar atmosphere. The $Rh_{NPs}$/AC catalyst was prepared by impregnation of $Rh_2(CO)_4Cl_2$ solution in dichloromethane with pure AC and then calcined at 673 K for 2 h under an Ar atmosphere. The theoretical loading of Rh for all the catalysts was 1 wt%.

**Catalyst performance evaluation**
The direct oxidation of ethane was carried out in a 100 mL batch reactor containing a PTFE lining. For a typical experiment, 50 mg catalyst and 10 g water were added, followed by purging the reactor five times with

2.0 MPa ethane (99.999%). Three megapascal ethane was then added at room temperature firstly, followed by 1.0 MPa CO and 0.5 MPa oxygen. After no variation in pressure, the autoclave was heated to the reaction temperature at a rotational speed of 600 rpm and maintained for a certain time. After the reaction, the reactor was rapidly moved into an ice bath and lowered to below 283 K. The reaction time refers to the thermostatic period. The gas phase products analysis was performed on Agilent Technologies 7890B gas chromatography system using a TDX-01 packed column. The liquid-phase product was obtained by suction filtration of the obtained liquid-phase mixture. Two-hundredths percentage by weight DSS solution was prepared by dissolving 4,4-dimethyl-4-silapentane-1-sulfonic acid (DSS) in deuterium oxide ($D_2O$) as the internal standard. A linear relationship between the peak areas ratio of oxygenates to DSS was used to establish a standard curve, setting the

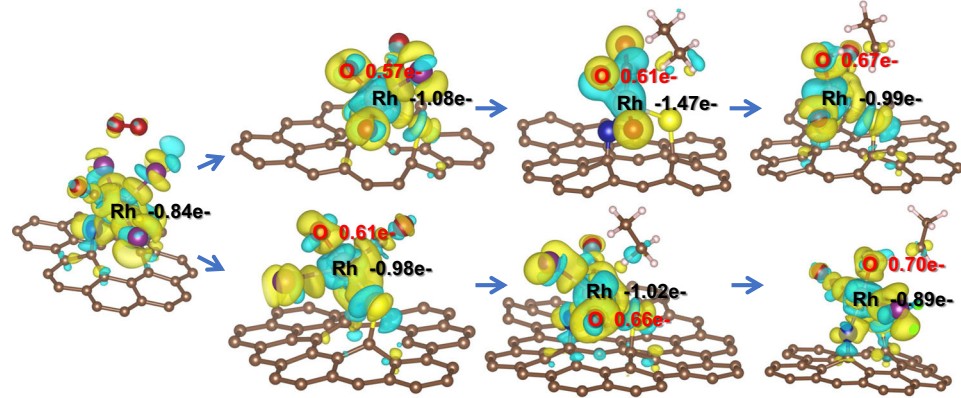

**Fig. 5 | DFT calculations.** The differential charge density on the Rh active site in both pathways in units of negative charge $e^-$. The Bader charge change values for Rh and O are marked. For the differential charge density, the cyan areas are electron deficient and the yellow areas are electron rich.

DSS chemical shift as $\delta = 0$. 1400 μL liquid product and 200 μL DSS standard solution were mixed up by ultrasonic for 10 min. Eight hundred microliter of the mixed liquid was extracted for the $^1$H-NMR test. $^1$H NMR was collected on a Bruker 700 MHz spectrometer. According to the qualitative analysis method by NMR results, the chemical shifts at $\delta = 1.17, 3.65$ ppm were attributed to $CH_3CH_2OH$; the chemical shifts at $\delta = 1.33, 2.23, 9.67$ ppm were attributed to $CH_3CHO$, and the chemical shift at $\delta = 2.08$ ppm was attributed to $CH_3COOH$. Each data was essentially repeated over three times under the same terms. The carbon balance was between 95% and 105% for the catalytic tests conducted. The acetaldehyde selectivity (%), and TOF were calculated as following Eqs. (1–2):

$$\text{acetaldehyde selectivity} = \frac{n_{(\text{acetaldehyde})}}{n_{(\text{oxygenates})}} \times 100\% \qquad (1)$$

$$\text{TOF} = \frac{n_{(\text{oxygenates})}}{n_{(\text{Rh})} \times t} \qquad (2)$$

## Catalyst characterization

SEM images were obtained with a Quanta 400 FEG instrument at an accelerating voltage of 0.01–30 kV. A Quanta 400 FEG instrument was used to obtain SEM images at an acceleration voltage of 0.01–30 kV. HR-TEM images were taken with a TECNAI G2 F30 instrument at an accelerating voltage of 200 kV. The high-angle circular dark-field scanning transmission electron microscope (HAADF-STEM) and EDS mapping images were obtained on a JEM-ARM200F instrument with a CEOS probe corrector working at 200 kV to guarantee a resolution of 0.08 nm. The Quantachrome Autosorb-1 system was used to determine the nitrogen sorption isotherms at the temperature of liquid nitrogen and the samples were outgassed for 12 h at 393 K before the measurements. The pore size distribution curves of the adsorbed branch were determined by the quenched solid density functional theory (QSDFT) method. Within the relative pressure $P/P_0 = 0.3$, the specific surface area was calculated from the adsorption data using the Brunauer–Emmett–Teller (BET) method. The pore size distribution of AC and the catalyst was mainly concentrated in the micropore range (<2 nm) calculated from the QSDFT adsorption method. XRD measurements were performed using a PANalytical X'Pert Pro X-ray diffractometer with a Cu Kα X-ray source at a wavelength of 1.5045 Å. CO-TPD MS of $Rh_1$/AC-SNI was performed by an AMI-300 chemisorption analyzer. Sixty milligram sample was first treated for 1 h at 303 K in a flow of He (30 mL min$^{-1}$) to remove physically adsorbed substances. After the baseline stabilized, TPD data was collected using a thermal conductivity detector (TCD) at a heating rate from 10 K min$^{-1}$ to 1273 K. To quantify the total desorbed CO, the peak area of the TCD signal was

calibrated by pulse sampling using a 10% CO/He standard gas. XPS was performed with a Thermo Fisher Scientific ESCALAB 250Xi instrument under Al Kα irradiation, and the binding energy was calibrated with reference to the C 1 s peak (284.8 eV). The inductively coupled plasma emission spectrometer (ICP-OES) analysis was performed on the PerkinElmer Optima 7300 DV instrument, and the powder samples were pretreated on the Anton Paar Multiwave 3000 system.

The $C_2H_6$–$D_2$-TPD-MS experiment was carried out on an Autochem II 2920 instrument. $D_2$ was used as the carrier gas and the flow rate was 15 mL min$^{-1}$. Ethane (99%) was used as the pulse gas. Fifty milligram sample was firstly pretreated at 373 K under an Ar atmosphere for 30 min and then switched to a $D_2$ atmosphere. After the baseline was stabilized, the temperature was raised to 423 K and held for 1 h with 20 pulses of ethane, and meanwhile monitored by mass spectrometry. The EPR spectrums were obtained on a Bruker A200 instrument with a microwave frequency of 9.3246 GHz. The reaction autoclave was quickly transferred to an ice bath and cooled to below 278 K after the reaction, then the liquid phase product was extracted and quickly transferred to a cold tank stored below 273 K by liquid nitrogen. The gas chromatography-mass spectrometry (GC-MS) experiment was carried out on a gas chromatography linked with quadrupole battle-time high-resolution mass spectrometer (Agilent Technologies Inc 8890-7250), which was equipped with a DB-WAX column and two 5A mol sieve column, TCD, and flame ionization detector (FID). The column temperature was kept at 313 K for 5 min, then increased to 523 K at a rate of 10 K min$^{-1}$, and then kept at 523 K for 10 min.

The short-lived intermediate species can be observed via in situ FEL-TOF/MS experiments were conducted to detect the in situ formed short-lived radical species. For in situ experiments, the $Rh_1$/AC-SNI sample was placed at the back end of the quartz tube reactor. The rear end was designed as an inverted cone with an extremely small hole at the apex for the diffusion of reactants and radical species into the detection chamber, and the back end of the quartz tube was wrapped with resistance wire to heat the reaction tube. After introducing $C_2H_6$/CO/$O_2$/$H_2O$/Ar into the reactor, the pressure of the reactor and chamber was kept at $10^{-2}$ and $10^{-7}$ Torr. $H_2O$ steam was carried in through $C_2H_6$/CO/ Ar bubbling at ambient temperature. Subsequently, the $Rh_1$/AC-SNI sample was heated from room temperature to the test temperature in the $C_2H_6$/CO/$O_2$/$H_2O$/Ar mixture flow, ($C_2H_6$/CO = 1:1, 5 mL min$^{-1}$; $O_2$/Ar = 1:1, 5 mL min$^{-1}$). Substances in the detection chamber were ionized by the VUV FEL and the resulting ions were analyzed by time-of-flight mass spectrometry equipped with a microchannel plate detector. In this work, the pulse energy was 8 - 9 μJ/pulse operating at 10 Hz, and the output wavelength was continuously 115 nm. The EXAFS spectrum of the catalyst was tested on the edge of Rh K using the Si (3 1 1) crystal monochromator at the BL14W1 beamline of SSRF of SINAP (Shanghai, China). The storage ring operated at 3.5 GeV

and the injection current was 200 mA. Rh foil was used as a reference sample, and all X-ray absorption spectra were obtained in fluorescence mode.

## Density functional theory (DFT) calculations

All the DFT calculations were performed using Vienna Ab-initio Simulation Package (VASP) code[53,54] with electron correction treated within the generalized gradient approximation using the Perdew–Burke–Ernzerhof (PBE) exchange-correlation functional[55,56]. The projector augmented wave (PAW) method[57] was used to treat the effect of the inner cores on the valence states. The slab of Rh anchored on the N/S co-doped carbon materials was set. The reaction on the surface of single-Rh-sites (co-doped C (001)) was carried out using the slab models composed of $p(7 \times 7)$ supercells with a single layer. In all the calculations, the cutoff energy was set to be 500 eV and the Gaussian electron smearing method with $\sigma = 0.05$ eV was used. The convergence tolerance for residual force and energy on each atom during structure relaxation was set to 0.05 eV/Å and $10^{-5}$ eV, respectively. The Monkhorst–Pack grids[58] were set to be $4 \times 4 \times 1$ surface optimizations. A vacuum layer of 20 Å along the $z$ direction was introduced to eliminate the spurious interactions between adjacent sheets.

## Data availability

All data supporting the findings of this study are available within the paper and its Supplementary Information files.

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

## Acknowledgements

The financial support of this work by the CAS Project for Young Scientists in Basic Research (YSBR-022), the National Natural Science Foundation of China (22372161, 22002156, and 22108275), the National Key R&D Program of China, (No. 2023YFA1506604), the Strategic Priority Research Program of the Chinese Academy of Sciences (No. XDA29040400), the Innovation Program for Quantum Science and Technology (No. 2021ZD0303305), the Youth Innovation Promotion Association of the Chinese Academy of Sciences (2022179, 2021181). We also acknowledge the Shanghai Synchrotron Radiation Facility (SSRF) for providing the beam time for the XAFS experiments. We thank the Institute Center For Shared Technology and Facilities of DICP, CAS for the help in using GC–MS and $C_2H_6$–$D_2$-TPD-MS.

## Author contributions

These authors contributed equally: Bin Li, Jiali Mu, Guifa Long. Bin Li, Jiali Mu, Ende Huang, Siyue Liu, and Yao Wei performed the experiments under the supervision of Xiangen Song, Li Yan, Fanfei Sun, Wenrui Dong, Weiqing Zhang, and Yunjie Ding, who conceptualized the research and acquired research funding for the project. Siquan Feng, Qiao Yuan, Yutong Cai, and Jian Song participated in various experiments and discussions. Guifa Long performed the theoretical calculations. Bin Li and Jiali Mu, wrote the paper with substantial input and revision from Xiangen Song, Yunjie Ding, and Xueming Yang.

## Competing interests

There are no competing interests.
