## [Peer Review File · Nature Communications]

Water-Participated Mild Oxidation of Ethane to AcetaldehydeREVIEWER COMMENTS

Reviewer #1 (Remarks to the Author):

This manuscript presents an investigation of ethane partial oxidation using S, N, I-modified carbon-supported atomically dispersed Rh catalysts.

I believe that the paper represents incremental progress in this area. The most important finding is the incorporation of oxygen from water into the products. However, because the catalyst system is so complex and not discussed much, it is not clear to the reviewer how this information can be used by the field. For example, can those working with well-defined sites in zeolites, which are known for hydrocarbons, benefit from this information?

I have additional minor comments to consider.

Can the authors explain in the introduction why S,N, and I -doped activated carbon is used? There was no context provided for why such a complex support material is needed.

Why does the XAS suggest that the charge of Rh δ^+ is between +1 and +3 if its white line intensity lies between the zero valent and +3 Rh standards?

The CO-TPD MS is said to confirm a monocarbonyl forms on the AC-SNI sample. Does it also confirm the dicarbonyl on the other support indicated?

It is necessary to explain at the beginning of the catalyst evaluation section what kind of reactor and reaction conditions were examined. When I get to this part of the paper, I still have no idea what the system is? Aqueous, batch, O₂ pressure? It is even unclear at this point if H₂O₂ is involved because that was discussed in the introduction. Why is CO introduced along with ethane and O₂? How is TOF calculated? What is the reaction temperature? All of this info should be in the text as well.

Reviewer #2 (Remarks to the Author):

In this manuscript, the authors prepared various single metals and metal clusters on activated carbon modified with N, S, and I and applied them to selective oxidize ethane into ethanol and/or acetaldehyde. Various characterization techniques were also conducted to analyze the catalysts before and after a reaction and to understand the reaction routes for each product. This manuscript is well-organized and well-written. However, the following issue should be properly resolved.

- The role of CO in the feed is not clear in this manuscript. As you might know, Sen and his co-workers have already published the selective oxidation of alkanes in the CO/O₂/H₂O system. They claimed that in situ generated H₂O₂ from this system was essential to activate C-H bond in alkanes. Does your system follow the similar pathway of them? If not, the role of CO should be clearly explained. If yes, the DFT calculation should be completely revised according to Sen's proposal.

Reference

J. Am. Chem. Soc. 1997, 119, 26, 6048–6053

**Dalian Institute of Chemical Physics, Chinese Academy of
Sciences**

Zhongshan Road 457, Dalian 116023, P.R. China.

29 February 2024

Responses to the Nature Communications

Manuscript ID: NCOMMS-23-55400A

Title: Water-Participated Mild Oxidation of Ethane to Acetaldehyde

Author(s): Bin Li, Jiali Mu, Guifa Long, Xiangen Song, Ende Huang, Siyue Liu, Yao Wei, Fanfei Sun, Siquan Feng, Qiao Yuan, Yutong Cai, Jian Song, Wenrui Dong, Weiqing Zhang, Xueming Yang, Li Yan, Yunjie Ding.

Dear editor and reviewers,

On behalf of the co-authors, we thank you very much for the thoughtful, constructive, and positive feedback, which is very useful for us to improve our manuscript and the future work. We have carefully considered the editor's and reviewers' advice and comments and tried our best to revise our manuscript. The reviewer comments are laid out below in *italicized font*. Our response is given in normal font with blue text. The modifications provided in the revised Manuscript are shown **with yellow highlight**.

Sincerely yours,

Prof. Yunjie Ding (Ph. D)

Dalian Institute of Chemical Physics, Chinese Academy of Sciences

457 Zhongshan Road, Dalian 116023, P. R. China

Email: dyj@dicp.ac.cn

Response to the reviewer's comments:

Reviewers 1 (Remarks to the author):

This manuscript presents an investigation of ethane partial oxidation using S, N, I-modified carbon-supported atomically dispersed Rh catalysts.

I believe that the paper represents incremental progress in this area. The most important finding is the incorporation of oxygen from water into the products. However, because the catalyst system is so complex and not discussed much, it is not clear to the reviewer how this information can be used by the field. For example, can those be working with well-defined sites in zeolites, which are known for hydrocarbons, benefit from this information?

Response: Thank you very much for raising this interesting discussion on the applicability of our catalyst system for this field. To elucidate this point, we designed a series of experiments to verify the assumption. Firstly, using H-ZSM-5 molecular sieve with a Si : Al ratio of 50 as the carrier, we prepared a 1wt% Rh-ZSM-5 catalyst using RhCl₃ as the precursor referring to the work of Maria and Tao et al^{1,2}. The Rh-ZSM-5 catalyst exhibits a TOF of 24.2 h⁻¹ as shown in **Supplementary Fig. 23**.

Supplementary Fig. 23 | Activity comparison of Rh₁/AC-SNI and Rh-ZSM-5.

Reaction conditions: T = 423 K, $P_{C_2H_6}$ = 3.0 MPa, P_{CO} = 1.0 MPa and P_{O_2} = 0.5 MPa for 2 h, m_{water} = 10 (g), $m_{catalyst}$ = 50 mg.

Moreover, we conducted ¹⁸O-GC-MS experiments for the products of Rh-ZSM-5

catalyst. The ^{18}O distribution of oxygenates for Rh-ZSM-5 is shown in **Supplementary Fig. 24**. Despite of the higher acetic acid selectivity on Rh-ZSM-5, the ^{18}O distribution of oxygenates for Rh-ZSM-5 is basically consistent with the results of Rh₁/AC-SNI. A large proportion of $\text{C}_2\text{H}_5^{18}\text{O}$ ($m/z=47$) was generated, showing that ethanol product was basically marked with ^{18}O (**Fig. 3a** and **Supplementary Fig. 24a**). Surprisingly, almost none of ^{18}O was detected in acetaldehyde (**Fig. 3b, 3c** and **Supplementary Fig. 24b, 24c**), indicating that O_2 only participated in the formation of ethanol. To further explore the oxygen sources of acetaldehyde, we used ^{18}O -labeled water for isotope experiments. **Fig. 3a** and **Supplementary Fig. 24a** showed that none of ethanol was labeled by ^{18}O from water. While all CH^{18}O ($m/z=31$), $\text{C}_2\text{H}_3^{18}\text{O}$ ($m/z=45$) and $\text{C}_2\text{H}_4^{18}\text{O}$ ($m/z=46$) fragments from acetaldehyde were labeled by ^{18}O from water (**Fig. 3b** and **Supplementary Fig. 24b**). As shown in **Fig. 3c** and **Supplementary Fig. 24c**, most of the O in acetic acid originated from H_2^{18}O , verified by a large amount of CH^{18}O_2 ($m/z=49$) and $\text{C}_2\text{H}_4^{18}\text{O}_2$ ($m/z=64$). This indicates that the low-temperature direct oxidation reaction mechanism of ethane proposed under this reaction condition has certain universality.

Rh-ZSM-5

Supplementary Fig. 24 | GC-MS spectra of (a) ethanol (b) acetaldehyde (c) acetic acid, respectively with $\text{H}_2\text{O}/^{18}\text{O}_2/\text{H}_2^{18}\text{O}$ isotope labeling experiment on Rh-ZSM-5 catalyst.

Fig. 3 | GC-MS spectra of (a) ethanol (b) acetaldehyde (c) acetic acid, respectively with H₂O/ D₂O/ ¹⁸O₂/ H₂¹⁸O isotope labeling experiment on Rh₁/AC-SNI catalyst.

I have additional minor comments to consider.

Comments: *Can the authors explain in the introduction why S, N, and I -doped activated carbon is used? There was no context provided for why such a complex support material is needed.*

Response: Thank you very much for your nice suggestion. In response to your question, we have added the following content to the introduction: Heterogeneous single-metal-site catalysts (HSMSCs) have recently emerged as an important class of high-efficiency catalysts with almost 100% atomic utilization and unique properties,^{3,4} which can ideally bridge the gap between heterogeneous and homogeneous catalysis and offer a platform for understanding the nature of active sites at the molecular level.⁵ The relationships between coordination atoms and metal center determine the electronic state and geometry of single-metal-site and further the performance for catalytic reaction. In recent years, the gradual development of tuning the local coordination environment of single-metal-sites by doping N atoms on carbon carriers has emerged^{6,7}. Additionally, sulfur as an electron-donating ligand can also form coordination bond with the active center, influencing electron cloud density and thus enhancing reaction activity^{8,9}. We previously revealed that S species can promote the activity of methanol carbonylation by coordination with single-Rh-site^{10,11}. Furthermore, the crucial role of

I species in achieving atomic dispersion of Rh nanoparticles was investigated^{3,12}. Introducing multifarious coordination atoms (S, N, I) onto the active carbon support as anchoring sites to anchor single-metal-site offers a prospective pathway for synthesizing HSMSCs. Herein, we first show that the single-Rh-site bound on S, N, I doped activated carbon (Rh₁/AC-SNI) in the form of Rh mononuclear complex efficiently catalyzes the direct conversion of ethane to C₂ oxygenate products using O₂ oxidizing agent at 423 K in aqueous solution.

Comments: *Why does the XAS suggest that the charge of Rh^{δ+} is between +1 and +3 if its white line intensity lies between the zero valent and +3 Rh standards?*

Response: Thank you very much for your careful review of our paper. The mismatch between the charge of Rh^{δ+} and the white line intensity is due to our typo. Accordingly, we have made changes to this part of the article: The X-ray absorption near-edge structure (XANES) spectra exhibits that all the absorption of Rh on Rh₁/AC-NI, Rh₁/AC-SI and Rh₁/AC-SNI is located between those of RhI₃ and Rh foil, revealing that the valence state of Rh^{δ+} species is between 0 and +3 (**Fig. 1e**).

Comments: *The CO-TPD MS is said to confirm a monocarbonyl forms on the AC-SNI sample. Does it also confirm the dicarbonyl on the other support indicated?*

Response: Thank you very much for your valuable comments. According to your suggestion, we also supplemented the CO-TPD experiments on the other two catalysts, and the experimental results also match with the EXAFS and DFT results. The specific description follows as: It can be found that the molar ratio of CO/Rh for Rh₁/AC-SNI, Rh₁/AC-NI and Rh₁/AC-SI are 1.06, 1.04 and 1.94, respectively, suggesting the coordination number of Rh-CO on Rh₁/AC-SNI, Rh₁/AC-NI and Rh₁/AC-SI were 1, 1 and 2, respectively (**Supplementary Fig. 9**).

Supplementary Fig. 9 | CO-TPD of Rh₁/AC-SNI, Rh₁/AC-SI and Rh₁/AC-NI.

Comments: *It is necessary to explain at the beginning of the catalyst evaluation section what kind of reactor and reaction conditions were examined. When I get to this part of the paper, I still have no idea what the system is? Aqueous, batch, O₂ pressure?*

Response: Thank you very much for your valuable comments. We are very sorry for your confusion and have made some adjustments in the article structure. We added a brief description of the reaction operation as “A typical experiment was carried out in a 100 mL batch reactor containing a polytetrafluoroethylene (PTFE) lining by adding 50 mg catalyst to 10 g deionized water under the conditions of 30 bar C₂H₆, 1 bar CO and 0.5 bar O₂ at 423K.”.

And a more detailed catalyst evaluation procedure has been placed in the catalyst evaluation section of the methods described as “The direct oxidation of ethane was carried out in a 100 mL batch reactor containing a polytetrafluoroethylene (PTFE) lining. For a typical experiment, 50 mg catalyst and 10 g water were added, followed by purging the reactor 5 times with 2.0 MPa ethane (99.999%). 3.0 MPa ethane was then added at room temperature firstly, followed with 1.0 MPa CO and 0.5 MPa oxygen. After no variation in pressure, the autoclave was heated to the reaction temperature at a rotational speed of 600 rpm and maintained for a certain time. After the reaction, the

reactor was rapidly moved into an ice bath and lowered to below 283 K. The reaction time referred to the thermostatic period. The gas phase products analysis was performed on Agilent Technologies 7890B gas chromatography system using a TDX-01 packed column. The liquid-phase product was obtained by suction filtration of the obtained liquid-phase mixture. 0.02wt% DSS solution was prepared by dissolving 4,4-dimethyl-4-silapentane-1-sulfonic acid (DSS) in deuterium oxide (D₂O) as the internal standard. A linear relationship between the peak areas ratio of oxygenates to DSS was used to establish a standard curve, setting the DSS chemical shift as $\delta=0$. 1400 μ L liquid product and 200 μ L DSS standard solution were mixed up by ultrasonic for 10 min. 800 μ L of the mixed liquid was extracted for ¹H-NMR test. ¹H NMR was collected on a Bruker 700MHz spectrometer. According to the qualitative analysis method by NMR results, the chemical shifts at $\delta=1.17, 3.65$ ppm were attributed to CH₃CH₂OH; the chemical shifts at $\delta=1.33, 2.23, 9.67$ ppm were attributed to CH₃CHO, and the chemical shift at $\delta= 2.08$ ppm was attributed to CH₃COOH.”.

Comments: *It is even unclear at this point of H₂O₂ is involved because that was discussed in the introduction.*

Response: Thank you very much for your valuable comments. Firstly, in the introduction, we describe those other studies used the H₂O₂ as oxidizing agent. H₂O₂ is not introduced in our system. Secondly, to address the concerns about whether H₂O₂ is generated in our system, we conduct some contrast experiment.

Colorimetric comparison method was used to detect the H₂O₂ intermediate species of the post-reaction solution using Ce(SO₄)₂ solution. The color of the mixture of Ce(SO₄)₂ solution and solution after reaction solution almost unchanged in contrast with that of Ce(SO₄)₂ solution (**Fig. R1**). The further UV-visible spectroscopy showed essentially consistent results of the two solutions. It indicated that H₂O₂ is not formed in our system (**Fig. R2**).

Fig. R1 | Color comparison of several solutions: (a) solution after reaction (sample), (b) 0.1mol/L $\text{Ce}(\text{SO}_4)_2$, (c) $\text{Ce}(\text{SO}_4)_2$ +sample, (d) 0.1mol/L $\text{Ce}(\text{SO}_4)_2$ and $\text{Ce}(\text{SO}_4)_2$ +sample.

Fig. R2 | UV spectra of the solution for $\text{Ce}(\text{SO}_4)_2$, $\text{Ce}(\text{SO}_4)_2$ +sample and $\text{Ce}(\text{SO}_4)_2$ + H_2O . The absorption in the 380-430 nm region indicates no formation of

H₂O₂.

Comments: *Why is CO introduced along with ethane and O₂?*

Response: Thank you for this interesting discussion. We would like to elucidate this question by the following points: 1. CO is a crucial role for enhancing the overall performance of this reaction. As displayed in the control experiment of **Fig 2c**, the TOF is close to 0 in the absence of CO, indicating that CO plays an irreplaceable role during the reaction process. 2. We conducted quasi in-situ XPS experiments to verify the role of CO, and the results have been incorporated into the manuscript described as: “Additionally, Maria and Ding Ma et al. believe that CO is an important ligand that can maintain the valence state of metals^{1,13}. We designed quasi in-situ XPS to analyze the role of CO on Rh₁/AC-SNI during the reaction. The sample was pretreated in a mixture of C₂H₆, H₂O and O₂ at 423 K for 2 h. The binding energy of Rh 3d 5/2 overall shifted to a higher value at 311.6 eV (**Supplementary Fig. 11** and **Supplementary Table 3**). This indicates that CO plays a role in maintaining the valence state of Rh during the reaction, ensuring the optimal coordination environment of active sites.”

Fig.2 | Ethane oxidation performance and structure-activity relationship. Catalytic activity of (c) different pressure of CO.

Supplementary Fig. 11 | XPS patterns of Rh 3d for fresh Rh₁/AC-SNI, spent Rh₁/AC-SNI and Rh₁/AC-SNI without CO atmosphere pretreatment.

Supplementary Table 3 | XPS results of Rh 3d for Rh/AC-I, Rh₁/AC-NI, Rh₁/AC-SI, fresh Rh₁/AC-SNI, spent Rh₁/AC-SNI and Rh₁/AC-SNI+C₂H₆+H₂O+O₂.

Catalyst	Rh ¹⁺		Rh ³⁺	
	B.E. (eV)	Area (%)	B.E. (eV)	Area (%)
Rh ₁ /AC-I	309.01	74.21%	310.71	25.79%
Rh ₁ /AC-SI	308.56	71.08%	310.80	28.92%
Rh ₁ /AC-NI	309.07	69.65%	310.97	30.35%
Rh ₁ /AC-SNI-fresh	308.97	65.76%	311.26	34.24%
Rh ₁ /AC-SNI-spent	308.92	63.65%	311.32	36.35%

Rh ₁ /AC-SNI +C ₂ H ₆ +H ₂ O+O ₂	309.10	60.16%	311.57	39.84%
--	--------	--------	--------	--------

Comments: *How is TOF calculated?*

Response: In accordance with your suggestions, we have incorporated the calculation formulas for TOF and acetaldehyde selectivity in the catalyst evaluation section of the methods, as detailed below: The acetaldehyde selectivity (%), and turnover frequency (TOF) were calculated as following equations (1-2):

$$\text{acetaldehyde selectivity} = \frac{n_{(\text{acetaldehyde})}}{n_{(\text{oxygenates})}} \times 100\% \quad (1)$$

$$\text{TOF} = \frac{n_{(\text{oxygenates})}}{n_{(\text{Rh})} \times t} \quad (2)$$

Comments: *What is the reaction temperature?*

Response: Thank you very much for your valuable comments. A typical experiment was carried out in a 100 mL batch reactor containing a polytetrafluoroethylene (PTFE) lining by adding 50 mg catalyst to 10 g deionized water under the conditions of 30 bar C₂H₆, 1 bar CO and 0.5 bar O₂ at **423K**. And the evaluation results regarding the change of reaction temperature have been given in **Fig 2e**.

Reviewers 2

In this manuscript, the authors prepared various single metals and metal clusters on activated carbon modified with N, S, and I and applied them to selective oxidize ethane into ethanol and/or acetaldehyde. Various characterization techniques were also conducted to analyze the catalysts before and after a reaction and to understand the reaction routes for each product. This manuscript is well-organized and well-written. However, the following issue should be properly resolved.

Response: We appreciate your positive comments and valuable suggestions to improve our manuscript. We have made a point-by-point response to your comments and carefully revised the manuscript as you suggested.

Comments: - *The role of CO in the feed is not clear in this manuscript. As you might know, Sen and his co-workers have already published the selective oxidation of alkanes in the CO/O₂/H₂O system. They claimed that in situ generated H₂O₂ from this system was essential to activate C-H bond in alkanes. Does your system follow the similar pathway of them? If not, the role of CO should be clearly explained. If yes, the DFT calculation should be completely revised according to Sen's proposal.*

Reference

J. Am. Chem. Soc. 1997, 119, 26, 6048 - 6053

Response: We sincerely appreciate your recognition of our work. Firstly, regarding about your question about whether H₂O₂ will be produced, the results of ¹⁸O-GC-MS have given some clue. O₂ only participated in the formation of ethanol. The oxygen sources of acetaldehyde came from water. If H₂O₂ is generated in the system, the oxygen in the products should all come from O₂.

Colorimetric comparison method was used to detect the H₂O₂ intermediate species of the post-reaction solution using Ce(SO₄)₂ solution. The color of the mixture of Ce(SO₄)₂ solution and solution after reaction solution almost unchanged in contrast with that of Ce(SO₄)₂ solution (**Fig. R1**). The further UV-visible spectroscopy showed essentially consistent results of the two solutions. It indicated that H₂O₂ is not formed in our system (**Fig. R2**).

Fig. R1 | Color comparison of several solutions: (a) solution after reaction (sample), (b) 0.1mol/L $\text{Ce}(\text{SO}_4)_2$, (c) $\text{Ce}(\text{SO}_4)_2$ +sample, (d) 0.1mol/L $\text{Ce}(\text{SO}_4)_2$ and $\text{Ce}(\text{SO}_4)_2$ +sample.

Fig. R2 | UV spectra of the solution for $\text{Ce}(\text{SO}_4)_2$, $\text{Ce}(\text{SO}_4)_2$ +sample and $\text{Ce}(\text{SO}_4)_2$ + H_2O . The absorption in the 380-430 nm region indicates no formation of H_2O_2 .

For the role of CO, we would like to elucidate this question by the following points:

1. CO is a crucial role for enhancing the overall performance of this reaction. As displayed in the control experiment of **Fig 2c**, the TOF is close to 0 in the absence of CO, indicating that CO plays an irreplaceable role during the reaction process.
2. We conducted quasi in-situ XPS experiments to verify the role of CO, and the results have been incorporated into the manuscript described as: “Additionally, Maria and Ding Ma et al. believe that CO is an important ligand that can maintain the valence state of metals^{1,13}. We designed quasi in-situ XPS to analyze the role of CO on Rh₁/AC-SNI during the reaction. The sample was pretreated in a mixture of C₂H₆, H₂O and O₂ at 423 K for 2 h. The binding energy of Rh 3d 5/2 overall shifted to a higher value at 311.6 eV (**Supplementary Fig. 11** and **Supplementary Table 3**). This indicates that CO plays a role in maintaining the valence state of Rh during the reaction, ensuring the optimal coordination environment of active sites.”

Fig.2 | Ethane oxidation performance and structure-activity relationship. Catalytic activity of (c) different pressure of CO.

Supplementary Fig. 11 | XPS patterns of Rh 3d for fresh Rh₁/AC-SNI, spent Rh₁/AC-SNI and Rh₁/AC-SNI without CO atmosphere pretreatment.

Supplementary Table 3 | XPS results of Rh 3d for Rh/AC-I, Rh₁/AC-NI, Rh₁/AC-SI, fresh Rh₁/AC-SNI, spent Rh₁/AC-SNI and Rh₁/AC-SNI+C₂H₆+H₂O+O₂.

Catalyst	Rh ¹⁺		Rh ³⁺	
	B.E. (eV)	Area (%)	B.E. (eV)	Area (%)
Rh ₁ /AC-I	309.01	74.21%	310.71	25.79%
Rh ₁ /AC-SI	308.56	71.08%	310.80	28.92%
Rh ₁ /AC-NI	309.07	69.65%	310.97	30.35%
Rh ₁ /AC-SNI-fresh	308.97	65.76%	311.26	34.24%
Rh ₁ /AC-SNI-spent	308.92	63.65%	311.32	36.35%

Rh ₁ /AC-SNI	309.10	60.16%	311.57	39.84%
+C ₂ H ₆ +H ₂ O+O ₂				

References:

- 1 Shan, J. J., Li, M. W., Allard, L. F., Lee, S. S. & Flytzani-Stephanopoulos, M. Mild oxidation of methane to methanol or acetic acid on supported isolated rhodium catalysts. *Nature* **551**, 605-606, (2017).
- 2 Tang, Y. *et al.* Single rhodium atoms anchored in micropores for efficient transformation of methane under mild conditions. *Nat. Commun.* **9**, 11, (2018).
- 3 Feng, S. *et al.* In situ formation of mononuclear complexes by reaction-induced atomic dispersion of supported noble metal nanoparticles. *Nat. Commun.* **10**, 5281, (2019).
- 4 Qi, J. *et al.* Selective methanol carbonylation to acetic acid on heterogeneous atomically dispersed ReO₄/SiO₂ Catalysts. *J. Am. Chem. Soc.* **142**, 14178-14189, (2020).
- 5 Cui, X., Li, W., Ryabchuk, P., Junge, K. & Beller, M. Bridging homogeneous and heterogeneous catalysis by heterogeneous single-metal-site catalysts. *Nat. Catal.* **1**, 385-397, (2018).
- 6 Kim, Y. T., Uruga, T. & Mitani, T. Formation of single Pt atoms on thiolated carbon nanotubes using a moderate and large-scale chemical approach. *Adv. Mater.* **18**, 2634-2638, (2006).
- 7 Kumar, P. *et al.* Multifunctional carbon nitride nanoarchitectures for catalysis. *Chem. Soc. Rev.* **52**, 7602-7664, (2023).
- 8 Wang, L. *et al.* A sulfur-tethering synthesis strategy toward high-loading atomically dispersed noble metal catalysts. *Sci. Adv.* **5**, (2019).
- 9 Yang, C. L. *et al.* Sulfur-anchoring synthesis of platinum intermetallic nanoparticle catalysts for fuel cells. *Science* **374**, 459-460, (2021).
- 10 Feng, S. *et al.* Sulfur-poisoning on Rh NP but sulfur-promotion on single-Rh₁-site for methanol carbonylation. *Appl. Catal. B-Environ.* **325**, 122318, (2023).
- 11 Mu, J. *et al.* Engineering the coordination environment of single-Rh-site with N and S atoms for efficient methanol carbonylation. *Adv. Funct. Mater.* **33**, 2305823, (2023).
- 12 Li, B. *et al.* Direct conversion of methane to oxygenates on porous organic polymers supported Rh mononuclear complex catalyst under mild conditions. *Appl. Catal. B-Environ.* **293**, (2021).
- 13 Jin, R. *et al.* Low temperature oxidation of ethane to oxygenates by oxygen over iridium-cluster catalysts. *J. Am. Chem. Soc.* **141**, 18921-18925, (2019).

REVIEWERS' COMMENTS

Reviewer #1 (Remarks to the Author):

The team has conducted additional clarifying experiments that I believe significantly strengthen the study, and answered my questions, and I have no further edits.

Reviewer #2 (Remarks to the Author):

In this revised manuscript, the authors have addressed most of the issues raised by the two reviewers based on additional experiments. Therefore, this reviewer recommends that this manuscript be accepted for publication in this journal.

Some additional questions I have is that when CO acts only on ligands of the Rh species, there is no noticeable oxidation of CO. Is this true? If oxygen in water is involved in the oxidation of ethane to acetaldehyde ($\text{C}_2\text{H}_6 + \text{H}_2\text{O} \rightarrow \text{CH}_3\text{CHO} + \text{H}_2$), then hydrogen should be produced. Have you found any H_2 production, and if not, what is the exact chemical reaction that synthesizes acetaldehyde from ethane?

**Dalian Institute of Chemical Physics, Chinese Academy of
Sciences**

Zhongshan Road 457, Dalian 116023, P.R. China.

29 February 2024

Responses to the Nature Communications

Manuscript ID: NCOMMS-23-55400A

Title: Water-Participated Mild Oxidation of Ethane to Acetaldehyde

Author(s): Bin Li, Jiali Mu, Guifa Long, Xiangen Song, Ende Huang, Siyue Liu, Yao Wei, Fanfei Sun, Siquan Feng, Qiao Yuan, Yutong Cai, Jian Song, Wenrui Dong, Weiqing Zhang, Xueming Yang, Li Yan, Yunjie Ding.

Dear editor and reviewers,

On behalf of the co-authors, we thank you very much for the thoughtful, constructive, and positive feedback, which is very useful for us to improve our manuscript and the future work. We have carefully considered the editor's and reviewers' advice and comments and tried our best to revise our manuscript. The reviewer comments are laid out below in *italicized font*. Our response is given in normal font with blue text.

Sincerely yours,

Prof. Yunjie Ding (Ph. D)

Dalian Institute of Chemical Physics, Chinese Academy of Sciences

457 Zhongshan Road, Dalian 116023, P. R. China

Email: dyj@dicp.ac.cn

Response to the reviewer's comments:

Reviewers 1 (Remarks to the author):

The team has conducted additional clarifying experiments that I believe significantly strengthen the study, and answered my questions, and I have no further edits.

Response: We appreciate your positive comments and valuable suggestions to improve our manuscript.

Reviewers 2

In this revised manuscript, the authors have addressed most of the issues raised by the two reviewers based on additional experiments. Therefore, this reviewer recommends that this manuscript be accepted for publication in this journal.

Response: We appreciate your positive comments and valuable suggestions to improve our manuscript. We have made a point-by-point response to your comments and carefully revised the manuscript as you suggested.

Comments: - *Some additional questions I have is that when CO acts only on ligands of the Rh species, there is no noticeable oxidation of CO. Is this true?*

Response: We sincerely appreciate your recognition of our work. In fact, CO will be oxidized to CO₂ in the presence of O₂, which was also mentioned in the work of Maria and Tao^{1,2}. This result was also found in our work, so we believe that subsequent work needs to find a more efficient ligand to replace CO.

Comments: - *If oxygen in water is involved in the oxidation of ethane to acetaldehyde (C₂H₆ + H₂O -> CH₃CHO + H₂), then hydrogen should be produced. Have you found any H₂ production, and if not, what is the exact chemical reaction that synthesizes acetaldehyde from ethane?*

Response: We sincerely appreciate your recognition of our work. C₂H₆ + H₂O -> CH₃CHO + H₂ is correct. We did find that a trace amount of H₂ was produced in the

gas phase. Due to the O₂ in the reaction system, the small amount of H₂ generated can quickly react with O₂ to form new H₂O molecules at 423K. However, since there is excess H₂O in the system and the amount of H₂ generated is very small, the H₂O originate from H₂ and O₂ is not enough to affect the reaction results.

References:

- 1 Shan, J. J., Li, M. W., Allard, L. F., Lee, S. S. & Flytzani-Stephanopoulos, M. Mild oxidation of methane to methanol or acetic acid on supported isolated rhodium catalysts. *Nature* **551**, 605-606.
- 2 Tang, Y. *et al.* Single rhodium atoms anchored in micropores for efficient transformation of methane under mild conditions. *Nat. Commun.* **9**, 11.